# High Production of Nattokinase via Fed-Batch Fermentation of the γ-PGA-Deficient Strain of *Bacillus licheniformis*

Xin Li [1], Jing Yang [1], Jun Liu [1,*], Xiaohui Zhang [1], Wei Wu [1], Dazhong Yan [1], Lihong Miao [1], Dongbo Cai [2], Xin Ma [2] and Shouwen Chen [2,3,*]

1. School of Life Science and Technology, Wuhan Polytechnic University, Wuhan 430023, China; lixin084@whpu.edu.cn (X.L.); yjing828@hotmail.com (J.Y.); zhangxiaohui2524@163.com (X.Z.); 1741216705go@gmail.com (W.W.); yandz6808@163.com (D.Y.); miaowhpu@126.com (L.M.)
2. State Key Laboratory of Biocatalysis and Enzyme Engineering, Environmental Microbial Technology Center of Hubei Province, College of Life Sciences, Hubei University, Wuhan 430062, China; caidongbo@hubu.edu.cn (D.C.); maxin@hubu.edu.cn (X.M.)
3. Wuhan Kanglida Biotechnology Co., Ltd., Wuhan 430075, China
* Correspondence: junliu85@163.com (J.L.); chenshouwen@hubu.edu.cn (S.C.)

**Abstract:** During the production of nattokinase (NK) by *Bacillus* species, certain by-products such as poly-γ-glutamic acid (γ-PGA) are simultaneously synthesized. The impact of γ-PGA synthesis on NK production remains unclear. In this study, we knocked out the *pgsC* gene, a component of the γ-PGA synthetase cluster (*pgsBCA*), and constructed a novel recombinant strain, *Bacillus licheniformis* BL11. Next, we compared the fed-batch fermentation profiles of BL11 and its parental strain BL10, conducted transcriptional analysis, and measured intracellular ATP content. We also optimized glucose-feeding strategies under varying oxygen supply conditions. Our results indicated that the utilization rates of glucose and soybean meal were both improved in the *pgsC*-deficient strain BL11, and NK activity was enhanced. Furthermore, the transcriptional levels of genes involved in glycolysis and the TCA cycle were relatively upregulated in BL11. The maximal NK activity reached 2522.2 FU/mL at 54 h of fermentation using a constant glucose-feeding rate of 5.0 g/(L·h) under high oxygen supply conditions. The newly developed recombinant strain *B. licheniformis* BL11, along with the optimized feeding strategy, shows promise for large-scale NK production.

**Keywords:** nattokinase production; γ-PGA deficiency; *Bacillus licheniformis*; fed-batch fermentation

## 1. Introduction

Nattokinase (NK, also known as subtilisin NAT) (EC 3.4.21.62) is a basic serine protease, found in Japanese natto and many other kinds of traditional fermented bean products in Asia. Comprising 275 amino acids, NK possesses a molecular weight of 27.7 kDa and an isoelectric point (pI) of 8.6, demonstrating significant enzymatic activity within the pH range of 5.5 to 9.0 [1]. The most attractive features of NK are its potent fibrinolytic and thrombolytic activity [2]. Empirical evidence confirmed the prophylactic and alleviating effects of NK on cardiovascular diseases, including possessing anti-thrombosis and fibrinolytic activities [3–5], anticoagulant [6], blood lipid-lowering, and anti-atherosclerotic [7,8] properties, and has been shown to prevent hypertension [9,10]. Compared with traditional thrombolytic drugs (urokinase, streptokinase, and tissue plasminogen activator), NK exhibits several advantages such as a relatively robust safety profile, convenient oral administration, easy absorption, high thrombolytic efficiency, and sustained effectiveness [11]. Consequently, NK has been acknowledged as a beneficial dietary supplement and a promising candidate for thrombolytic therapy.

*Bacillus* is recognized as a safer and more efficient NK producer in comparison to *Pseudomonas* sp. and marine creatures [12]. Previously, NK-producing *Bacillus* strains were mainly isolated from fermented food or the natural environment [13–16], yet the fibrinolytic

enzyme activities generated by these strains fell short of matching the requirements of industrial production. With the advent of genetic and protein engineering, *Bacillus* has been employed in recent years as a host strain to express the endogenous or exogenous *aprN* gene, which codes for NK [17,18]. The components involved in expression and transport have been further optimized [19–22]. Moreover, due to the development of recombinant strains, the liquid fermentation method is progressively substituting the solid method, thereby becoming the primary mode of NK production. However, it is noteworthy that the by-products synthesized by *Bacillus* may exert entirely different effects on NK production when the fermentation method is altered. Poly-γ-glutamic acid (γ-PGA), a polypeptide composed of D-and L-glutamic acid units, is the main component in natto mucus. Many researchers have believed that the γ-PGA produced by *Bacillus* during the solid-state fermentation process could stimulate the biosynthesis of NK, as the NK activity and the γ-PGA production peak concurrently [23–25]. Yet, it remains to be elucidated whether γ-PGA synthesis continues to exert a positive influence on NK production under the liquid fermentation method.

On the other hand, research focused on optimizing the medium composition and culture parameters has progressed rapidly in recent years in liquid fermentation, mainly including carbon and nitrogen sources of the initial medium [26–29], along with the optimum pH, dissolved oxygen, and temperature during the process [10,30]. However, these optimizations were conducted using batch fermentation rather than fed-batch fermentation. Although some studies have reported significantly higher NK activity in fed-batch fermentation with an optimal feeding strategy compared to batch fermentation [31–33], the final yields and activities of NK in these research still fell short of meeting the demands of industrial NK production.

In the previous work, a recombinant *Bacillus licheniformis* strain BL10 was constructed through the deletion of ten genes related to extracellular proteases and the optimization of signal peptides. This resulted in achieving 33.83 FU/mL of NK activity through liquid fermentation in a shake flask. In this study, a novel recombinant strain, *Bacillus licheniformis* BL11, was constructed by knocking out the *pgsC* gene in *B. licheniformis* BL10, the highly conserved sequence within the active site of the γ-PGA synthetase. Then, the profiles of fed-batch fermentations between BL10 and BL11 were compared. Subsequently, transcriptional analysis and measurement of intracellular ATP content were conducted to investigate the potential mechanism of the effects of γ-PGA synthesis deficiency on NK production. Finally, glucose-feeding strategies were optimized under varying oxygen supply conditions. This study aimed to construct a newly recombinant strain and develop an optimal feeding strategy for the highly efficient production of NK. The findings of this study may contribute to the potential application of industrial-scale NK production.

## 2. Materials and Methods

### 2.1. Bacterial Strains and Plasmids

The bacterial strains and plasmids used in this study are listed in Table 1. *B. licheniformis* BL10 (CCTCC M2014253) was employed as the original strain for recombinant strain construction. T2(2)-Ori was applied for gene knockout in *B. licheniformis*, and the NK expression vector was constructed based on pHY300PLK. *E. coli* DH5α served as the host strain for plasmid construction. The primers used for strain construction and RT-qPCR are listed in Table S1.

### 2.2. Media and Flask Culture Conditions

The Luria–Bertani (LB) medium was served as the basic medium or seed medium for bacterial growth, and the corresponding titer of antibiotic (50 μg/mL ampicillin, 25 μg/mL tetracycline, or 20 μg/mL kanamycin) was added into the medium. The medium for NK production contained 6% glucose, 3% soybean meal, 0.5% corn steep liquor, 1% $(NH_4)_2SO_4$, 1% $NaNO_3$, 0.15% $K_2HPO_4 \cdot 3H_2O$, 0.05% $MgSO_4 \cdot 7H_2O$, and 0.05% $CaCl_2 \cdot 2H_2O$. In fed-batch fermentations, the feeding solution contained a glucose concentration of 500 g/L. The

seed culture was cultivated in a 250 mL flask containing 50 mL LB medium, and maintained at 37 °C in a rotary shaker at 230 rpm for 12 h.

**Table 1.** The strains and plasmids used in this research.

| Strains and Plasmids | Relevant Characteristics | Source of Reference |
|---|---|---|
| Strains | | |
| *E. coli* DH5α | *sup*E44 Δ*lac*U169 (f 80 *lacZ*ΔM15) *hsd*R17 *rec*A1 *gyr*A96 *thi*1 *rel*A1 | Lab collection |
| *Bacillus licheniformis* WX-02 | Wide-type host strain (CCTCC M208065) | Lab collection |
| BL10 | WX-02(Δ*mpr*, Δ*vpr*, Δ*aprX*, Δ*epr*, Δ*bpr*, Δ*wpr*A, Δ*apr*E, Δ*bpr*A, Δ*hag*, Δ*amy*L) harboring pP43SacCNK (CCTCC M2014253) | [20] |
| BL11 | BL10(Δ*pgs*C) harboring pP43SacCNK | This study |
| Plasmids | | |
| pP43SacCNK | Plasmid pHY300PLK harboring P43 promoter, signal peptide of *Sac*C, gene *apr*N, and *amy*L terminator | [20] |
| T$_2$(2)-Ori | *E. coli-B. licheniformis* shuttle vector, for gene knockout | Lab collection |
| T2-pgsC | T2(2)-Ori derivative containing homologous arms of *pgs*C, to block poly-γ-glutamic acid synthesis | This study |

### 2.3. Construction of pgsC-Deficient Strain

The gene *pgs*C was deleted in *B. licheniformis* BL10 according to our previous literature [22], and the construction procedure was described briefly as follows. First, the upstream and downstream homogenous arms were amplified using the corresponding primers (Table S1), and fused through splicing overlap extension (SOE)-PCR. The fused fragment was inserted into T2(2)-Ori at the restriction sites *Xba*I/*Sac*I. Diagnostic PCR and DNA sequencing confirmed that the plasmid named T2-PgsC was constructed successfully.

Next, the plasmid T2-PgsC was electro-transferred into *B. licheniformis* BL10, and was verified with diagnostic PCR and plasmids extraction. The positive transformants were cultivated in the LB liquid medium with 20 μg/mL kanamycin at 45 °C to promote the single-cross transformation, and then incubated in the kanamycin-free medium at 37 °C for several generations. The kanamycin-sensitive colonies were picked and verified via diagnostic PCR, and DNA sequencing results confirmed that the *pgs*C-deficient strain (*B. licheniformis* BL11) was constructed successfully.

### 2.4. Fed-Batch Fermentations

The fed-batch fermentations were carried out in 5 L mechanically stirred fermenters (T&J Bio-engineering (Shanghai) Co., Ltd., Shanghai, China). The inoculation volume ratio was 3% (*v/v*), and the temperature was controlled at 37 °C. The initial pH was adjusted to $7.0 \pm 0.1$ using an aqueous ammonia solution (concentration of 6 M), and then during the whole process of fed-batch fermentation, aqueous ammonia, and hydrochloric acid solution (concentration of 6 M) were used to maintain the pH between 6.5 and 7.5. The agitation speed and aeration rate were set at 400 r/min and 1.5 vvm, respectively. Under normal oxygen supply conditions, the fermentation medium volume was 2.5 L, and the agitation speed and aeration rate remained constant throughout the process. Under high oxygen supply conditions, the fermentation medium volume was reduced to 2 L, and the agitation speed and aeration rate were gradually increased to 700 r/min and 3 vvm, respectively.

Glucose solution feeding commenced at 12 h when the glucose concentration in the broth fell below 10 g/L. Three glucose-feeding strategies were designed for this study: (1) a constant glucose-feeding strategy at a rate of 2.5 g/(L·h); (2) a constant glucose-feeding strategy at a rate of 5.0 g/(L·h); (3) a sufficient glucose-feeding strategy, which aimed to maintain a constant presence of residual glucose in the broth during the whole process by employing a continuous glucose feeding at a rate of 5.0 g/(L·h), as well as intermittently adding a certain amount of glucose solution.

### 2.5. Analytic Methods

Biomass was measured through the determination of the optical density at 600 nm. Glucose concentration was determined using an SBA-40C bioanalyzer (Academy of Sciences, Ji'nan, China) [34]. The content of soybean meal was monitored by measuring the dry weight of the deposit, which was described in the previous literature [35].

The concentration of γ-PGA, the concentrations of amino acids in broth, and the contents of amino acids from acid hydrolysis of soybean meal were all measured with an HPLC (Agilent Technologies 1260 series, Ankeny, IA, USA). A TSK Gel G6000 PWXL gel permeation chromatogram column (7.8 mm × 300 mm, Tosoh, Tokyo, Japan) was used to determine the γ-PGA yield; while the information about mobile phase, flow rate, wavelength, and the injecting volume was according to the previously reported method [36]. The concentrations of amino acids were determined by using o-phthaldialdehyde (OPA) pre-column derivatization. A Polaris 5 C18-A column (250 mm × 4.6 mm) was used, and the samples were detected at 338 nm with an ultraviolet (UV) detector. The elution program was performed according to the published method [37].

The NK activity was measured according to our previously reported method [22]. In brief, 0.4 mL fibrinogen solution (0.72%, *w/v*) and 1.4 mL Tris-HCl (50 mM, pH 7.8) were pre-incubated in a test tube at 37 °C for 10 min. Thereafter, 0.1 mL thrombin solution (20 U/mL) was added, followed by the addition of 0.1 mL diluted sample, and the mixture was incubated at 37 °C for 60 min. Finally, 2 mL trichloroacetic acid (TCA) solution (0.2 M) was added to stop the reaction, and the absorbance of the supernatant was determined at 275 nm after centrifugation (12,000× *g*, 10 min). One unit of NK activity (1 FU) was defined as the amount of enzyme leading to the 0.01 increase for A275 in 1 min.

The NK protein concentration in the broth was determined according to the method reported in the literature [38]. The mixture of 900 μL broth and 100 μL TCA solution (6.1 M) was maintained at 4 °C overnight, and then was centrifuged at 12,000× *g* for 10 min. The precipitate was washed three times by using absolute alcohol, dried and redissolved in a solution containing 2.0 M thiourea and 8.0 M urea. The redissolved sample was mixed with an equal volume of 2× SDS-PAGE loading buffer, and 10 μL of the mixtures was subjected to SDS-PAGE analysis. Bovine serum albumin (BSA) was used as the standard, and the protein concentration of the sample was estimated using the software QUANTITY ONE (version 4.6.2) loaded in GS-800 calibrated densitometer Bio-Rad (Hercules, CA, USA).

The intracellular ATP content was measured as follows: Two centrifuge tubes, each containing 1 mL of fermentation broth and 4 mL of distilled water, were centrifuged at 300× *g* for 10 min at 4 °C. The supernatant was then further centrifuged at 12,000× *g* for 10 min at 4 °C. The cell pellets in one tube were harvested and dried at 80 °C to measure the cell dry weight; while those in the other tube were resuspended by adding 200 μL of lysate buffer. The intracellular ATP concentration was determined based on the user manual of the ATP Assay Kit (Beyotime Biotechnology Co., Ltd., Shanghai, China), and then converted into ATP content per cell dry weight.

All samples were analyzed in triplicate, and the data were presented as the mean ± the standard deviation for each sample point.

### 2.6. Transcriptional Analysis

The cells of BL10 and BL11 were sampled at the fermentation period of 24 h and 36 h. RT-qPCR was conducted according to the previous research [39]. Briefly, the total RNA was extracted with TRIzol® Reagent (Thermo Fisher Scientific, Waltham, MA, USA), and then RNase-free DNase I enzyme (TaKaRa, Kusatsu, Japan) was used to digest trace DNA. RevertAid First Strand cDNA Synthesis Kit (Thermo, USA) was used to amplify the first stand of cDNA. The real-time PCR was performed with SYBR® Select Master Mix (ABI, Foster City, CA, USA) according to the manufacturer's instructions. The transcriptional levels for genes in the new recombinant strains (BL11) were compared with those of control strains (BL10) after normalizing to the reference gene 16S rRNA. All the samples were measured in triplicates, and the data were presented as the mean ± the standard deviation.

## 3. Results

### 3.1. Deficiency of γ-PGA Synthesis and Its Implication on NK Production

The γ-PGA synthetase is reportedly encoded by the *pgsBCAE* gene cluster wherein the *pgsC* gene is responsible for encoding the highly conserved sequence within the active site of the γ-PGA synthetase [40,41]. In this study, the *pgsC* gene in *B. licheniformis* BL10 was deleted by using the homologous recombination method. The PCR validation results, as shown in Figure 1, include the upstream and downstream homologous arms, the recombinant plasmid T2-pgsC, and the *pgsC* gene deletion. Finally, a new recombinant strain *B. licheniformis* BL11 was constructed successfully.

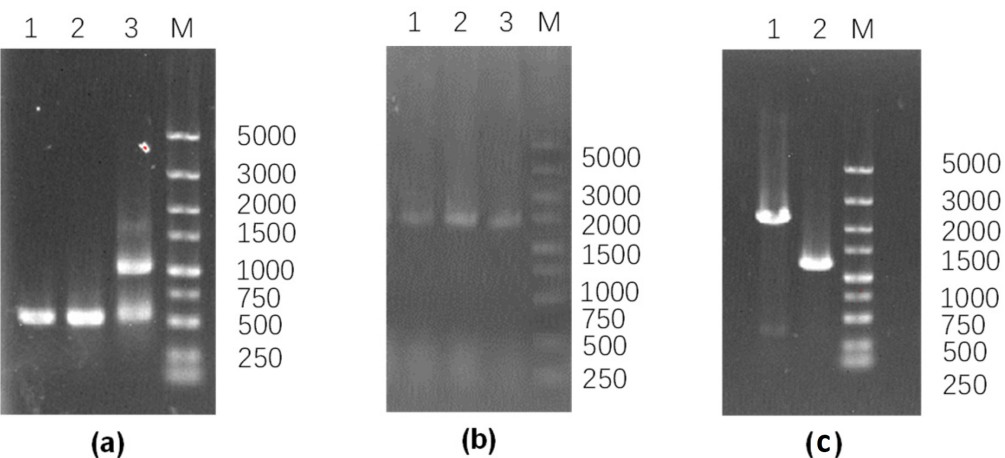

**Figure 1.** PCR amplification of knockouts and colony PCR verification. (**a**) PCR products of the homologous arms and the splicing fragment; Lane 1: PCR product of the upstream homologous arm of *pgsC* (541 bp), Lane 2: PCR product of the downstream homologous arm of *pgsC* (539 bp), Lane 3: SOE-PCR product of the upstream and the downstream homologous arm of *pgsC* (1044 bp), Lane M: DL5000 DNA Marker; (**b**) PCR products of DH5α/T2-pgsC. Lanes 1–3: PCR products of DH5α/T2-pgsC; (**c**) Confirmation of the *pgsC*-deficient strain; Lane 1: colony PCR of the control strain BL10 (2290 bp), Lane 2: colony PCR of the *pgsC*-deficient strain BL11 (1225 bp).

Next, fed-batch fermentations were carried out, and the fermentation profiles of BL10 and BL11 were compared to evaluate the effect of γ-PGA synthesis deficiency on NK production. As illustrated in Figure 2a, the biomass of the recombinant strain BL10 peaked at an OD600 value of 24.0 at 18 h, and then slightly declined during the subsequent fermentation process. NK activity reached the maximum of 377.8 FU/mL at 36 h, followed by a gradual decrease. γ-PGA production increased sharply after 36 h and achieved the maximal production of 8.5 g/L at the end. Concurrently, broth viscosity escalated in tandem with γ-PGA synthesis, reaching 660 mPa·s at the end of fermentation. In contrast, γ-PGA was undetectable in the fermentation broth of BL11, and the viscosity remained at a low level of 77 mPa·s, as depicted in Figure 2b. The biomass of BL11 peaked at an OD600 value of 30.7 at 54 h. NK activity steadily increased from 12 h, reaching the peak value of 880.9 FU/mL at 48 h, and then maintained above 800 FU/mL until the end. The maximal and final activity of NK produced by BL11 exhibited increments of 133.2% and 175.6%, respectively, compared to those produced by BL10. Therefore, the aforementioned results suggest that γ-PGA synthesis had a negative effect on NK production, while the deficiency of γ-PGA synthesis, achieved by knocking out the *pgsC* gene, could improve strain growth and promote NK production.

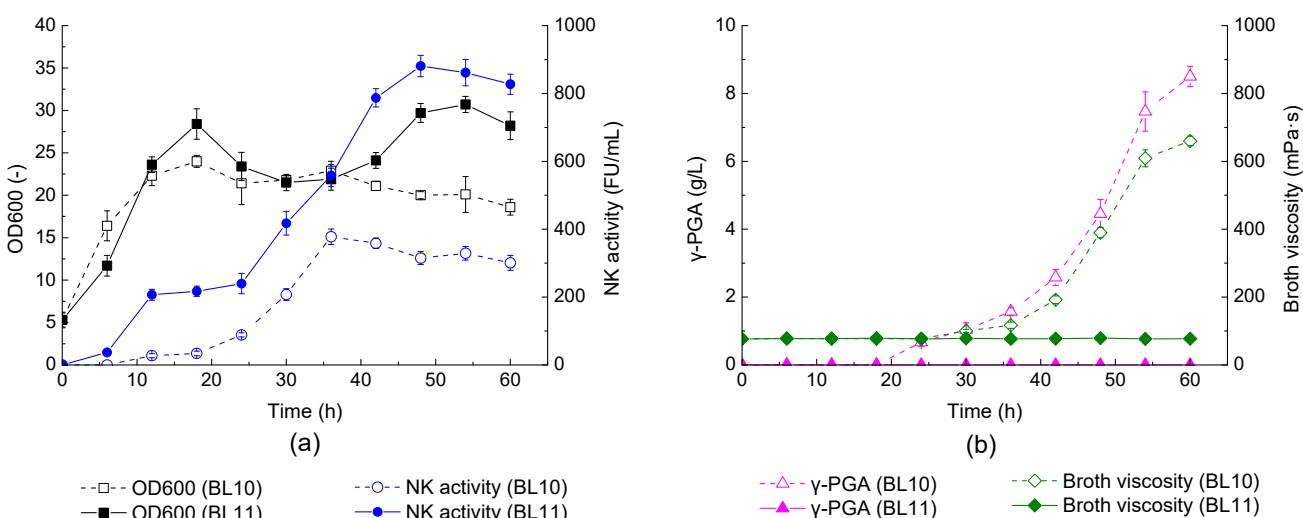

**Figure 2.** The profiles comparison of fed-batch fermentations between BL10 and BL11. (**a**) Comparison of OD600 and NK activity; (**b**) Comparison of γ-PGA and broth viscosity.

### 3.2. pgsC Deficiency Enhances the Utilization of Glucose and Soybean Meals

Glucose and soybean meals were used as the primary carbon and nitrogen sources in the fermentation medium, and their consumption was probably correlated with NK production. To evaluate the maximum total glucose consumption, the strains of BL10 and BL11 were both cultured via fed-batch fermentations using the sufficient glucose-feeding strategy. As depicted in Figure 3, the total glucose consumption of BL11 was 350 g/L, a 26.8% increase compared to that of BL10 (276 g/L). It was observed that the glucose consumption of BL10 was marginally lower than that of BL11 during the initial 36 h of fermentation. However, the BL10 strain exhibited a notable decline in its glucose consumption rate after 36 h, resulting in a significant disparity compared to that of BL11. On the other hand, the consumption rates of soybean meal also showed no obvious difference between BL10 and BL11 during the first 36 h. However, an unexpected increase in soybean meal concentration was observed in the fermentation of BL10 after 36 h, coinciding with a sharp increase in γ-PGA production and broth viscosity during the same fermentation period. Therefore, it could be inferred that the γ-PGA produced by BL10 may adhere to soybean meal, as well as other insoluble substances such as microbial cells, thereby potentially affecting the measurement accuracy of soybean meal concentration.

Because of the inaccurate measurement of soybean meal concentration in the BL10 broth, we subsequently investigated the concentrations of various amino acids in the broth at different fermentation times, as well as the contents of amino acids in the acid hydrolysates of soybean meal. The acid hydrolysate of soybean meal contained the highest contents of six amino acids, namely glutamate, alanine, aspartate, phenylalanine, lysine, and leucine (Table S2). During the middle and later stages of fermentation, the concentrations of four kinds of amino acids in BL11 broth were evidently lower than each of those in BL10, including glutamate, alanine, aspartate, and leucine, as shown in Figure 4. Moreover, the total amino acid concentration in BL11 broth reached 871.9 mg/L—a 37.4% decrease compared to that in BL10 broth (1393.4 mg/L). Therefore, these results indirectly demonstrated that the BL11 strain had a higher utilization rate of soybean meal compared to the BL10 strain, particularly during the middle and late stages of fermentation.

Additionally, the NK contents in the fermentation broth were also estimated, as shown in Figure 3c. The NK yield of BL11 was approximately 1.24 g/L at the end of fermentation, which was 3.4-fold higher than that of BL10 (0.36 g/L). This result was consistent with the NK activity assay result. Taken together, the deficiency of *pgsC* enhanced the utilization of both glucose and soybean meals, especially during the middle and late stages of fermentation, which further facilitated NK production.

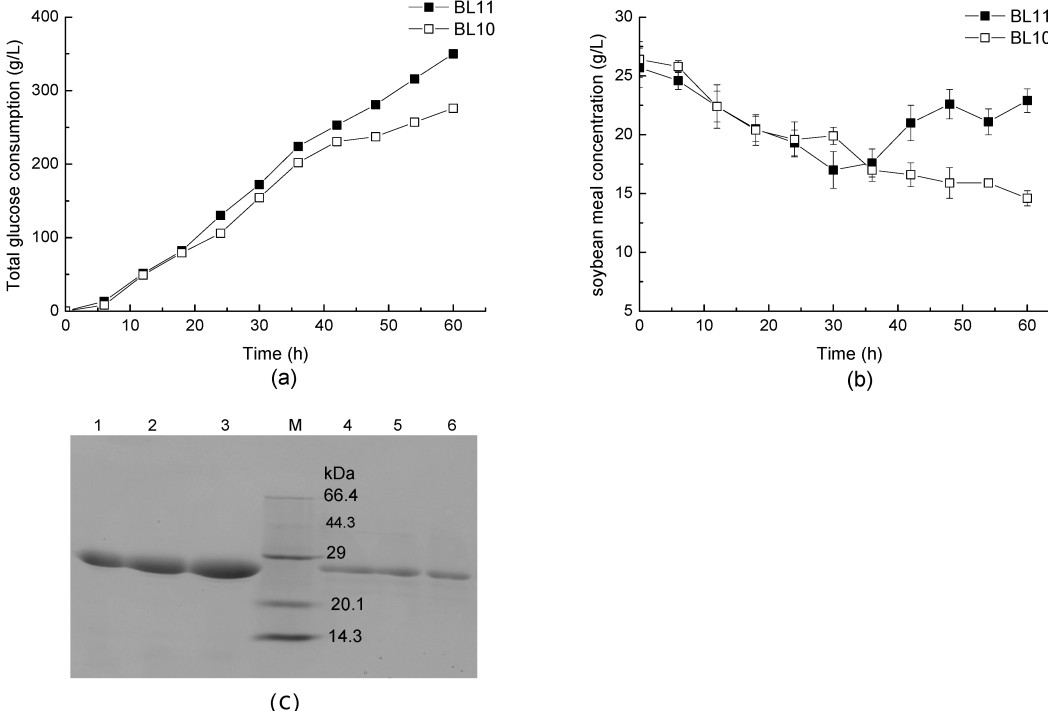

**Figure 3.** Effects of *pgsC* deficiency on the utilization rates of substrates and NK production. (**a**) The total glucose consumption by different strains; (**b**) The concentration of soybean meal in cultured broth of different strains; (**c**) SDS-PAGE analysis of NK production from different strains at different time points; Lane 1: BL11 at 36 h, Lane 2: BL11 at 48 h, Lane 3: BL11 at 60 h, Lane M: Premixed Protein Marker, Lane 4: BL10 at 60 h, Lane 5: BL10 at 48 h, Lane 6: BL10 at 36 h.

### 3.3. Effects of pgsC Deficiency on the Transcriptional Levels of Genes Involved in Glycolysis and Tricarboxylic acid Cycle, and ATP Content

In order to investigate the mechanism that *pgsC* deficiency enhances substrates utilization and NK production, the transcription levels of key genes involved in glycolysis, the tricarboxylic acid cycle (TCA cycle), and γ-PGA synthesis were determined, and the *aprN* gene carried by the plasmid for encoding the natto kinase synthase was also measured, as shown in Figure 5.

6-phosphofructokinase (encoded by *pfkA*) and pyruvate kinase (encoded by *pyk*) are generally regarded as the rate-limiting enzymes in the glycolysis of *Bacillus* [42], while citrate synthase (encoded by *citZ*), isocitric dehydrogenase (encoded by *icd*), and ketoglutaric dehydrogenase (encoded by *odhAB*) are the rate-limiting enzymes in the TCA cycle [43]. At the early stage of fermentation (24 h), the transcriptional levels of *pfkA* and *citZ* in BL11 were almost the same as those in BL10, and the transcriptional levels of *pyk*, *icd* and *odhA* were slightly increased by 0.7-fold, 2.1-fold, and 2.2-fold compared to those in BL10. By contrast, the transcriptional levels of *pfkA*, *pyk*, *icd*, and *odhA* in BL11 were significantly increased by 11.0-fold, 4.1-fold, 5.7-fold, and 11.3-fold, respectively, compared to those in BL10. This result provides a reasonable explanation for why BL11 exhibited higher efficiency of glucose and soybean meal utilization after 36 h. The transcriptional level of *pgsC* was essentially close to zero in BL11, which proved again that the *pgsC* gene was successfully knocked out. Even though none of γ-PGA was produced by BL11, the transcriptional levels of genes related to glutamate synthesis (*rocG* and *gltA*) were still upregulated at 36 h, which also reflected that the TCA cycle was strengthened. Moreover, the expression level of *aprN* in BL11 was nearly equal to that in BL10, which indicated that the difference in NK production between BL10 and BL11 was not caused by the differential expression of *aprN*.

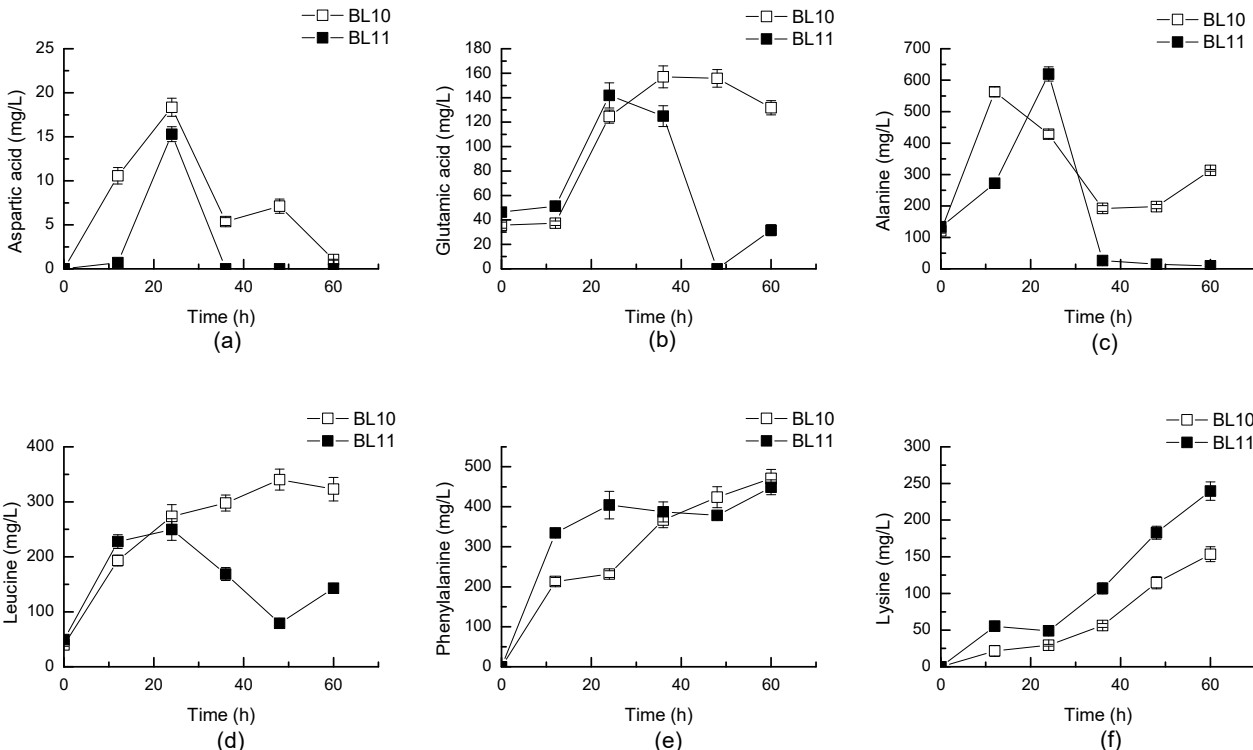

**Figure 4.** The concentrations of amino acids in cultured broth of different strains. (**a**) The concentration of aspartic acid; (**b**) The concentration of glutamic acid; (**c**) The concentration of alanine; (**d**) The concentration of leucine; (**e**) The concentration of phenylalanine; (**f**) The concentration of lysine.

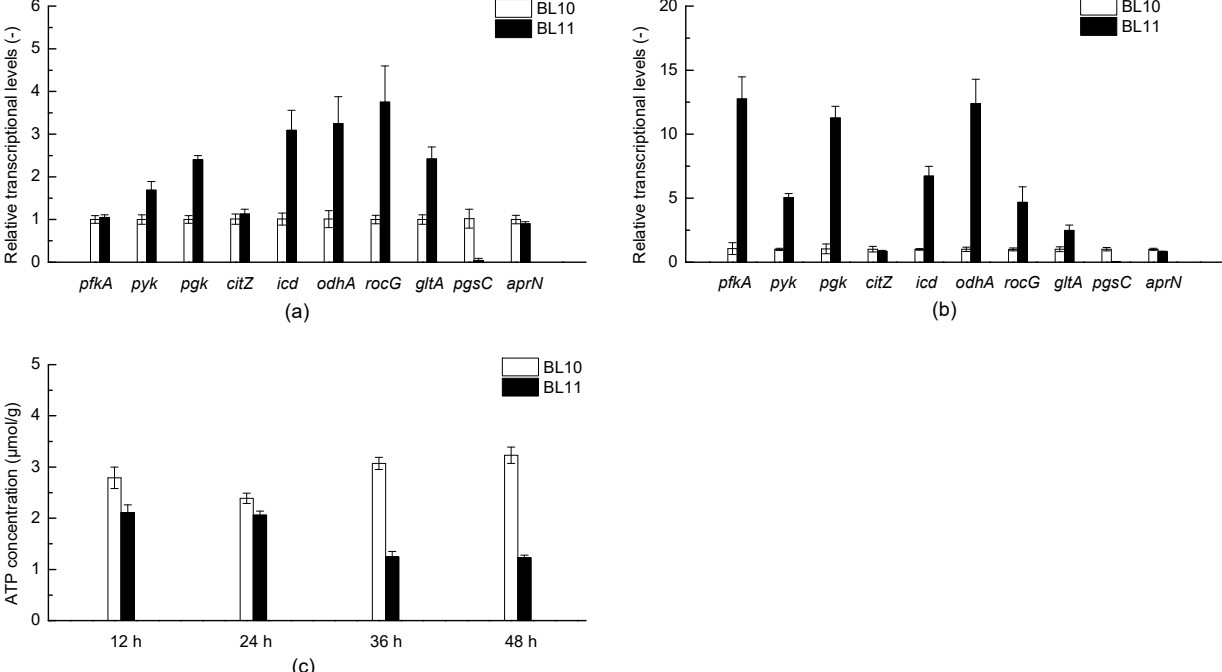

**Figure 5.** The transcriptional levels of genes involved in glycolysis and tricarboxylic acid cycle, and ATP content in different strains. (**a**) The transcriptional levels of key genes in BL10 and BL11 at 24 h; (**b**) The transcriptional levels of key genes in BL10 and BL11 at 36 h; (**c**) The ATP content in BL10 and BL11.

In addition, the catalytic reactions mediated by phosphoglycerate kinase (encoded by *pgk*) and pyruvate kinase in glycolysis are associated with ATP synthesis. Here, the transcriptional level of *pgk* was also elevated by 9.8-fold in BL11 compared to that in BL10 at 36 h. Moreover, the above results indicated that the glycolysis and TCA cycle in BL11 were more active than those in BL10, which suggested that more ATP should be synthesized in BL11. However, Figure 5c showed that the intracellular ATP concentration decreased greatly to 1.25 μmol/g in BL11 at 36 h, and then maintained at a low level. Comparatively, the ATP concentration increased slightly to 3.07 μmol/g in BL10 at 36 h and kept at this high level until the end. It is known that ATP is essential both for the heterologous protein expression and γ-PGA synthesis. A reasonable explanation was that the co-production of NK and γ-PGA exhibited a reciprocal inhibitory effect, resulting in reduced yields of both products and the accumulation of ATP in BL10. Conversely, the new recombinant strain BL11, with the deficiency of the *pgsC* gene, was incapable of synthesizing γ-PGA, leading to a rapid production of NK after 36 h followed by a significant consumption of ATP. Based on the above results, it could be concluded that the enhanced utilization of glucose and soybean meals in BL11 was achieved internally by upregulating the transcription levels of key genes involved in glycolysis and TCA cycle, as well as promoting ATP consumption.

### 3.4. Improving NK Production from BL11 via an Optimal Glucose-Feeding Strategy under Different Oxygen Supply Conditions

In this work, the fed-batch fermentations with three glucose-feeding strategies were conducted firstly under normal oxygen supply conditions, including the sufficient glucose-feeding strategy, and two other glucose-feeding strategies with constant rates of 2.5 and 5.0 g-glucose/(L·h). The performances of each fermentation are shown in Figure 6.

When the sufficient glucose-feeding strategy was adopted, the total glucose consumption was the highest among the three fed-batch fermentations, reaching 332.7 g/L. However, the maximum biomass and the soybean meal consumption were only 35.1 and 10.7 g/L, and the NK activity reached 831.1 FU/mL, which were all the lowest among the three fed-batch fermentations, as shown in Figure 6a. Next, in the fermentation by using the constant glucose-feeding strategy with the rate of 5.0 g/(L·h), the glucose concentration reached almost zero for a long time after the initial glucose was used up, which suggested that the feeding glucose was insufficient to match metabolic demands of the strain. The total glucose consumption was reduced to 287.7 g/L. By contrast, the maximum biomass, the soybean meal consumption, and the NK activity reached 42.6, 11.6 g/L, and 1067.5 FU/mL, respectively—21.4%, 8.4%, and 28.4 increases compared to each of those by using the sufficient glucose-feeding strategy. Further, when the constant rate of glucose feeding was set at 2.5 g/(L·h), the total glucose consumption was reduced correspondingly to 174.7 g/L. The maximum biomass reached 40.7, slightly lower than that using the constant rate of 5.0 g/(L·h). However, the soybean meal consumption and the NK activity reached 16.4 g/L and 1234.3 FU/mL, which were both the highest among the three fed-batch fermentations. Moreover, the NK yields of fermentation using the constant feeding rate of 2.5 g/(L·h) were further evaluated through SDS-PAGE analysis (Figure 6d), and the maximum yield reached about 1.68 g/L.

Subsequently, the three strategies of glucose feeding were used again under high oxygen supply conditions. The maximum biomass reached between 60 and 80 in these three fed-batch fermentations under high oxygen supply conditions, which was much higher than those under normal oxygen supply conditions. The total glucose consumption was 334.0 g/L in the fermentation using the sufficient glucose-feeding strategy under high oxygen supply conditions, almost the same as that under normal oxygen conditions; while the soybean meal consumption reached 17.4 g/L, a 62.6% increase compared to that under normal oxygen conditions. The maximum NK activity reached 1587.2 FU/mL at 48 h, an increment of 85.9% compared to that under normal oxygen supply conditions. However, this activity was still the lowest among the three fermentations conducted under high oxygen supply conditions. When the constant glucose-feeding strategy was implemented

at a rate of 2.5 g/(L·h), the consumption of soybean meal and the NK activity reached 20.9 g/L and 2019.6 FU/mL, respectively, which increased 20.1% and 27.2% compared to that using the sufficient glucose-feeding strategy under high oxygen supply conditions. Noticeably, the highest soybean meal consumption and NK activity were both observed in the fermentation using the constant rate of 5.0 g-glucose/(L·h), which reached 21.2 g/L and 2522.2 FU/mL, respectively. Additionally, the maximum yield of NK reached about 3.02 g/L by using the optimal feeding strategy under high oxygen supply conditions (Figure 7d). Based on it, the specific activity of NK was calculated to be 819.2 FU/mg. Therefore, the results above demonstrated that the sufficient glucose-feeding strategy was not beneficial to NK production, regardless of whether under high or normal oxygen supply conditions; by contrast, a slightly glucose-limited condition brought by the appropriate constant glucose-feeding strategy could promote the consumption of soybean meal, thereby resulting in the enhancement of NK production.

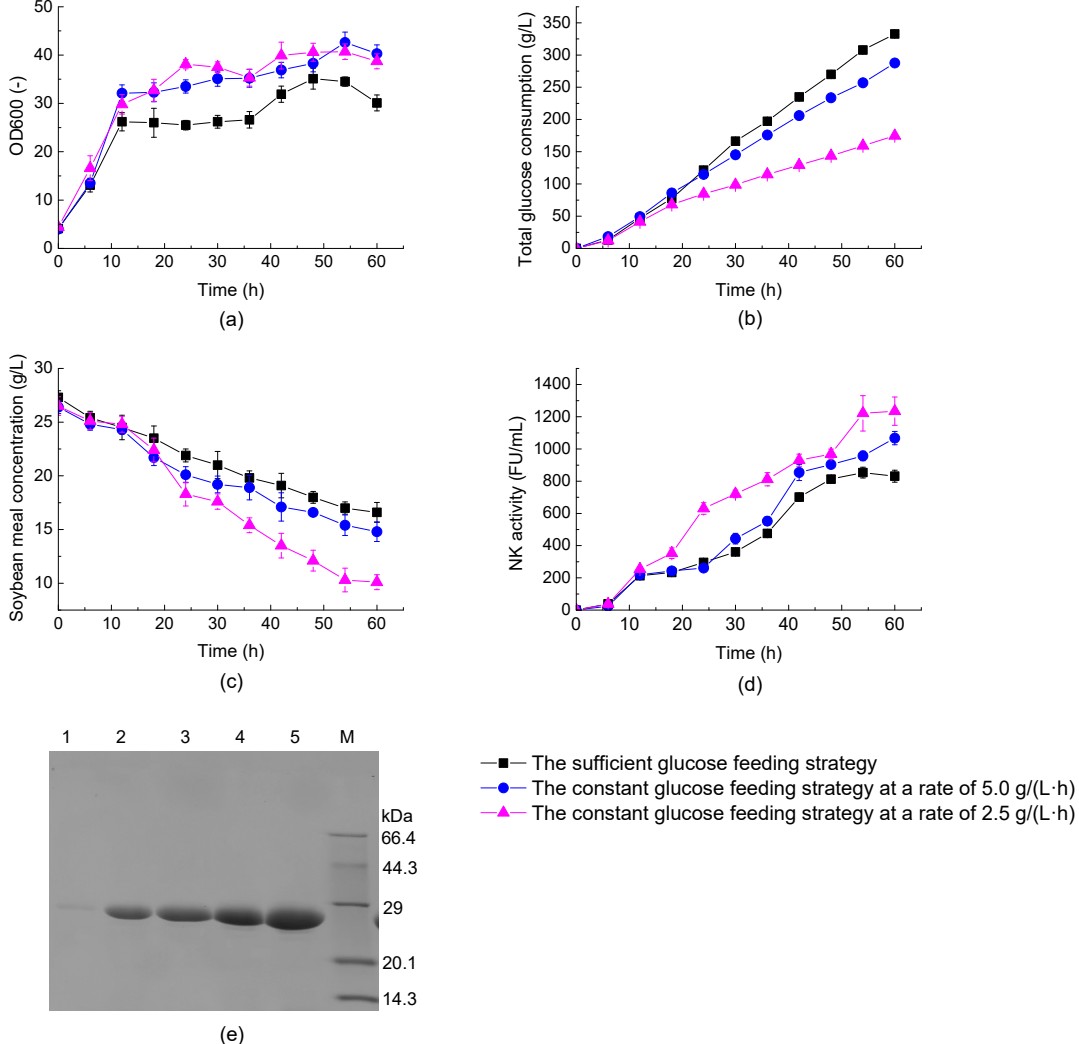

**Figure 6.** The profiles of fed-batch fermentations using different glucose-feeding strategies under normal oxygen supply conditions. (**a**) Comparison of OD600; (**b**) Comparison of total glucose consumption; (**c**) Comparison of soybean meal concentration; (**d**) Comparison of NK activity; (**e**) SDS-PAGE analysis of NK production at different time points in the fermentation with a constant glucose-feeding strategy at a rate of 2.5 g/(L·h); Lane 1: the sample at 12 h, Lane 2: the sample at 24 h, Lane 3: the sample at 36 h, Lane 4: the sample at 48 h, Lane 5: the sample at 60 h, Lane M: Premixed Protein Marker.

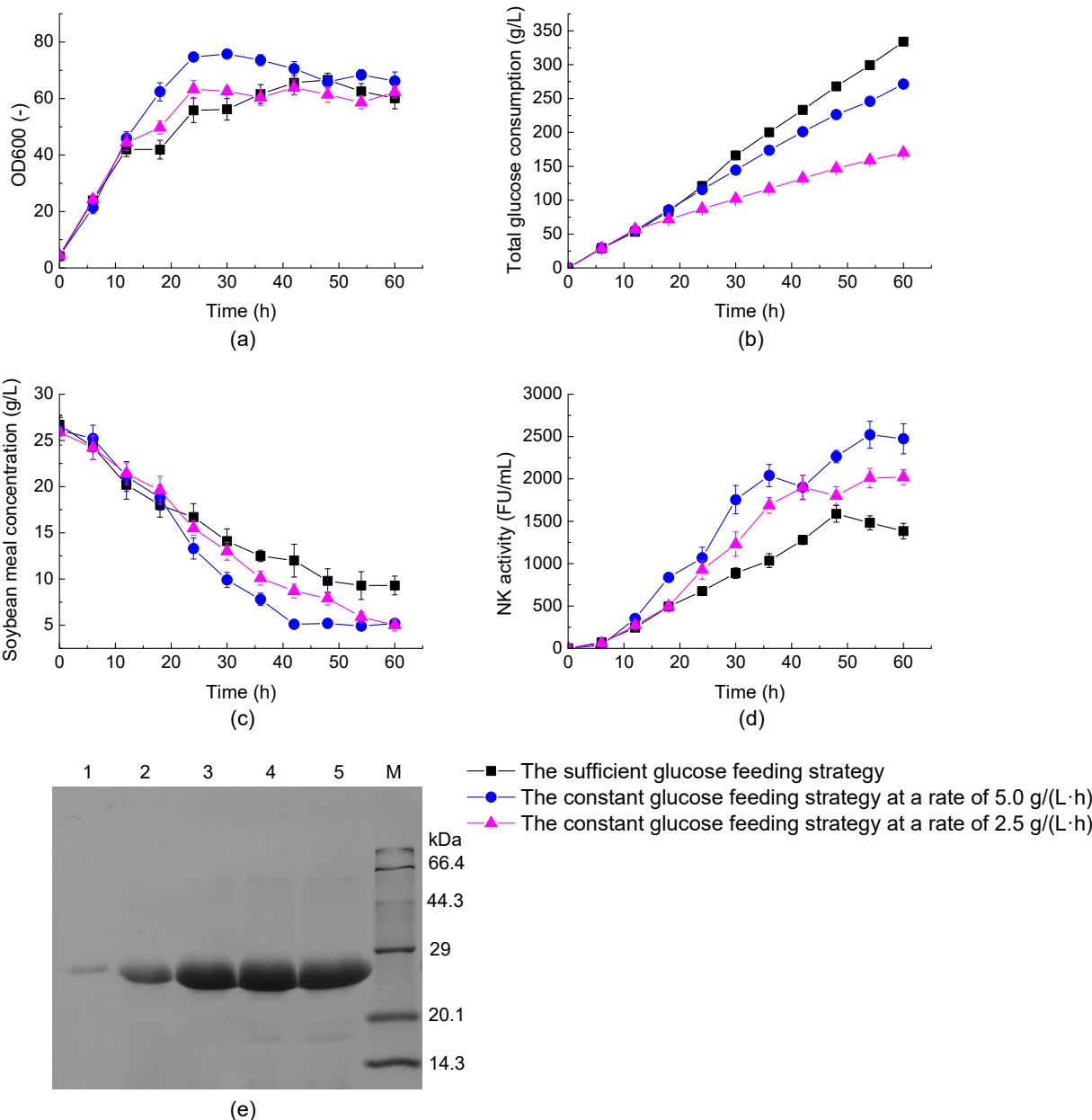

**Figure 7.** The profiles of fed-batch fermentations using different glucose-feeding strategies under high oxygen supply conditions. (**a**) Comparison of OD600; (**b**) Comparison of total glucose consumption; (**c**) Comparison of soybean meal concentration; (**d**) Comparison of NK activity; (**e**) SDS-PAGE analysis of NK production at different time points in the fermentation with a constant glucose-feeding strategy at a rate of 5.0 g/(L·h); Lane 1: the sample at 12 h, Lane 2: the sample at 24 h, Lane 3: the sample at 36 h, Lane 4: the sample at 48 h, Lane 5: the sample at 60 h, Lane M: Premixed Protein Marker.

## 4. Discussion

*Bacillus* expression system (*B. subtilis*, *B. amyloliquefaciens*, *B. licheniformis*, etc.) is commonly used for the efficient production of industrial enzymes, due to its good biological safety, superior capability of protein secretion, etc., and has been applied in the production of alkaline protease, neutral protease, α-amylase, β-mannanase, etc. [44]. Currently, many approaches have been used to enhance the activities or yields of NK in *Bacillus* species, as shown in Table 2, including optimization of promoters and signal peptides [18], over-expression or deficiency of transcription factors [22,35], and deletion of protease genes [20]. In fact, some by-products produced in the process of synthesizing NK probably have some

uncertain effects. γ-PGA, a by-product synthesized during NK production, has been widely reported as a great bonding carrier for NK [45,46], and the optimal conditions in the solid-state fermentation were explored for the co-production of NK and γ-PGA [24,25]. In this study, a rapid increase in γ-PGA production was observed after 36 h of BL10 fermentation, accompanied by the obvious decrease in the utilization rates of substrates and NK activity. On the one hand, the high broth viscosity induced by γ-PGA synthesis could severely hinder the utilization of substrates, thereby resulting in low metabolic activity of strains. The same phenomenon also appeared in our previous research on γ-PGA fermentation [47]. On the other hand, the γ-PGA synthesis would compete with the NK production for substrates, which could be another reason for the lower activity of NK in the BL10 broth. Li et al. have reported that NK production in a γ-PGA-producing strain may inhibit γ-PGA synthesis by alleviating cell hunger and strengthening substrate competition [48]. Upon comprehensive analysis of fermentation profiles and transcription level data, we found that the synthesis of γ-PGA did not increase the total glucose consumption, nor did it enhance the metabolism of the TCA cycle and glutamate synthesis. On the contrary, it brought a decline in glucose utilization and weakened the metabolic activity of the bacteria. Based on this, we believe that the high viscosity induced by γ-PGA synthesis, rather than substrate competition, is the primary factor triggering the above phenomena. Comparatively, a significant improvement in utilization rates of substrates and NK activity was achieved in a newly recombinant strain BL11, which lost the capability of γ-PGA synthesis by knocking out the *pgsC* gene.

It is known that substrates rich in legume proteins have great effects as the main nitrogen sources on NK yield and activities [49]. Increasing the concentrations of nitrogen sources has been proven in many research as an effective method to enhance the NK yield [50,51]. However, the substantial addition of nitrogen sources, particularly organic nitrogen sources, would increase substrate costs, and make the purification process more difficult. In comparison, enhancing the utilization rates of substrates may be a more efficient and economical approach for the improvement of NK production. In this study, the utilization rate of soybean meal was greatly improved in the *pgsC*-deficient strain BL11, as well as the consumption of free amino acids. It was also discovered that the transcriptional levels of key genes involved in glycolysis and the TCA cycle were mostly upregulated in BL11. Therefore, this *pgsC*-deficient strain would be a usable host to efficiently express protein/peptide for its enhancement in substrate utilization.

The natto fermentation was traditionally carried out using solid-state fermentation, and so was the NK production. Although solid-state fermentation of NK has advantages in cost and energy consumption, some obvious defects make it unsuitable for large-scale production, including poor fluidity, difficulties in the detection of intermediate parameters, and challenges in product extraction [12]. Comparatively, liquid fermentation is a more suitable method for large-scale NK production, particularly by using recombinant Bacillus strains. Most related research on the liquid fermentation of NK focused on optimizing components of the fermentation medium [26,50,52–54] and the main environmental factors [30,55]. However, fewer studies aimed at developing an optimal feeding strategy. In this study, we investigated the effects of different glucose-feeding strategies on NK production under both normal and high oxygen supply conditions. The results indicated that a slightly glucose-limited condition brought by the appropriate constant glucose-feeding strategy could promote the consumption of soybean meal, thereby resulting in the enhancement of NK production. The maximal NK activity achieved 2522.2 FU/mL in the fermentation using the constant feeding rate of 5.0 g-glucose/(L·h) under high oxygen supply conditions, a 2.86-fold higher than that before the optimization.

**Table 2.** Enhancement of nattokinase activity or yield through genetic engineering and optimization of fermentation conditions.

| Strains | Genetic Engineering Strategies | Culture Medium/Fermentation Substrates | Fermentation Process | Final Activity/Yield | References |
|---|---|---|---|---|---|
| *Bacillus subtilis* PSP2 | Heterologous expression of *aprN* gene | 1% trypsin, 1% oyster protein hydrolysate, 2% maltose, and 0.5% NaCl | Batch fermentation in a flask | 390.23 FU/mL | [17] |
| *Bacillus subtilis* WB800 | Optimization of promoters and signal peptides | 2.4% yeast extract, 2% tryptone, 0.4% glycerol, 2.3% $KH_2PO_4$, and 1.3% $K_2HPO_4$ | Batch fermentation in a flask | 292 FU/mL | [18] |
| *Bacillus licheniformis* DW2 | Deficiency of *dltABCD*, and expression of plasmid pP43SacCNK | 2% glucose, 1% peptone, 1% soy peptone, 1.5% yeast extract, 0.5% corn steep liquor, 0.6% $(NH_4)_2SO_4$, 0.3% $K_2HPO_4 \cdot 3H_2O$, and 1% NaCl | Batch fermentation in a flask | 38.02 FU/mL | [19] |
| *Bacillus licheniformis* BL10 | Deletion of protease genes, and expression of plasmid pP43SNT | Same as above | Batch fermentation in a flask | 33.83 FU/mL | [20] |
| *Bacillus licheniformis* BL10 | Overexpression of *sppA* in BL10 | Same as above | Batch fermentation in a flask | 45.05 FU/mL | [22] |
| *Bacillus licheniformis* DW2 | Deficiency of *bacA* and *aprA*, and expression of plasmid pP43SacCNK | 2% glucose, 1% peptone, 1% soy peptone, 1.5% yeast extract, 0.5% corn steep liquor, 0.3% $K_2HPO_4$, 0.6% $(NH_4)_2SO_4$, and 1% NaCl | Batch fermentation in a flask | 38.43 FU/mL | [35] |
| *Bacillus licheniformis* BL11 | Deficiency of *pgsC* in BL10 | 6% glucose, 3% soybean meal, 0.5% corn steep liquor, 1% $(NH_4)_2SO_4$, 1% $NaNO_3$, 0.15% $K_2HPO_4 \cdot 3H_2O$, 0.05% $MgSO_4 \cdot 7H_2O$, and 0.05% $CaCl_2 \cdot 2H_2O$ | Fed-batch fermentation in a 5 L fermenter | 2522.2 FU/mL | This study |

## 5. Conclusions

This study illustrated that γ-PGA synthesis had negative effects on NK production, and the deficiency of the *pgsC* gene indeed strengthened the metabolism of glycolysis and TCA cycle and promoted ATP consumption, leading to improvement in soybean meal consumption and NK activity. Then, the glucose-feeding strategies and the oxygen supply conditions were optimized, and the maximal NK activity reached 2522.2 FU/mL at 54 h in the fermentation with the constant glucose feeding of 5.0 g/(L·h) under high oxygen supply conditions. The newly developed recombinant strain *B. licheniformis* BL11, along with the optimized fermentation conditions, hold promise for large-scale production of NK.

**Supplementary Materials:** The following supporting information can be downloaded at https://www.mdpi.com/article/10.3390/fermentation9121018/s1, Table S1: Primers used for PCR and qPCR in this study; Table S2: The concentrations of amino acids in the acid hydrolysate of soybean meal.

**Author Contributions:** Conceptualization, X.L., J.L. and X.M.; methodology, X.L., J.Y., X.Z., W.W. and D.C.; software, X.Z. and W.W.; validation, J.L., X.M. and S.C.; formal analysis, D.Y.; investigation, D.Y. and D.C.; resources, L.M.; data curation, X.L.; writing—original draft preparation, X.L.; writing—review and editing, J.L. and X.M.; visualization, J.Y. and D.C.; supervision, J.L. and X.M.; project administration, X.L. and D.C.; funding acquisition, X.L., L.M. and X.M. All authors have read and agreed to the published version of the manuscript.

**Funding:** This research was funded by the National Key Research and Development Program of China (2021YFC2100202); the Key Research and Development Program of Hubei Province (2022BCA075,

2022BBA0031); and Open Project Funding of the State Key Laboratory of Biocatalysis and Enzyme Engineering (SKLBEE2022024).

**Institutional Review Board Statement:** Not applicable.

**Informed Consent Statement:** Not applicable.

**Data Availability Statement:** Data are contained within the article.

**Conflicts of Interest:** Author Shouwen Chen was employed by the company Wuhan Kanglida Biotechnology Co. The remaining authors declare that the research was conducted in the absence of any commercial or financial relationships that could be construed as a potential conflict of interest.

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
