# Peer review of "High Production of Nattokinase via Fed-Batch Fermentation of the γ-PGA-Deficient Strain of Bacillus licheniformis"

_fermentation, doi:10.3390/fermentation9121018_

Round 1

Reviewer 1 Report

Comments and Suggestions for Authors

  The manuscript entitled  "High production of nattokinase by fed-batch fermentation of the γ-PGA-deficient strain of Bacillus licheniformis", by Xin Li, Jing Yang, Jun Liu, Xiaohui Zhang, Wei Wu, Dazhong Yan, Lihong Miao, Dongbo Cai, Xin Ma and Shouwen Chen, describes the improvement of production of nattokinase production by the newly developed recombinant strain Bacillus licheniformis BL11, along with the optimized fermentation conditions.

  The concept of the work, the experimental procedure, and the interpretation of the results, described in this manuscript are sound. Several concerns are as follows;

L. 37;  Anti-thrombosis ---> anti-thrombosis

L. 68;  although ---> Although

L. 86;  Bacterial ---> Bacteria or bacterial strains

L. 95;  Luriae-Bertani ---> Luria-Bertani

L. 99;  K2HPO4 --->  K2HPO4

L. 151-;  The method to determine NK activity should be described in more detail. According to the reference cited [22],

  "The nattokinase activity was measured according to our previous reported method [8]: the mixture of 0.4 mL fibrinogen solution (0.72%, w/v) and 1.4 mL Tris–HCl (50 mM, pH 7.8) was incubated in a test tube at 37 °C for 10 min, followed by adding 0.1 mL thrombin solution (20 U/mL) to form the fibrin at 37 °C for 10 min. Then, the diluted sample solution (D) at the volume of 0.1 mL was added into the tube and shaken at 37 °C for 60 min at the interval of every 15 min, followed by adding 2 mL trichloroacetic acid (TCA) solution (0.2 M) to stop the reaction (AT). As for the control group, the mixture of 0.1 mL sample solution and 2 mL TCA solution (0.2 M) was used after incubating at 37 °C for 60 min (AB). Finally, all mixtures were centrifuged at 12000g for 10 min, and the absorbance of the supernatant was determined at 275 nm. One unit of nattokinase activity (1 FU) was defined as the amount of enzyme leading to the 0.01 increase for A275 in 1 min, and the formula was as follows: ". 

  Please show the essence in one sentence.

L.152;  "The concentrations of total extracellular proteins" are determined according to ref. [38], but the results or derived parameters such as specific activity of NK in any sample are not shown.

  The NK activity is always expressed in FU/mL.  There is no description of specific activity of NK (in FU/mg protein) nor description about the purity of NK in the culture media. Please state the content of NK in the cultur media, for example, the specific activity corresponding to 2522.2 FU/mL (in L. 420).  Description of the specific activity of purified NKase is also informative.

L. 173;  PRC ---> PCR

L. 180 & 186;  A600 value  --->  OD600

L. 242-243, 351-354, & 359-362;

  Judging from the SDS-PAGE, the culture media contained almost pure NK.  What kind of samples were spplied to SDS-PAGE --- were they the centrifuged supernatants of the culture media ? --- were their protein concentrations determined?  Please specify the conditions of SDS-PAGE in "analytical methods".

L. 251-254;  " In order to investigate the mechanism that pgsC deficiency enhances substrates utilization and NK production, the transcription levels of key genes involved in glycolysis, the tricarboxylic acid cycle (TCA cycle), and γ-PGA synthesis were determined, and the aprN gene carried by the plasmid for encoding the natto kinase synthase was also measured, as shown in Figure 5."

  This is confusing.  BL10 contains endogebeous NKgene but BL11 contain additional NKgene conferred exogeneously.  The effect of pgsC deficiency may be simply compared between WX02 and NKgene deficient mutant of WX02.  It seems more simple than using exogenouus NKgene.

L. 279-;  ATP concentration is sown in μmol/g.  Does the "g" mean the dry weight of the biomass?  Please specify the weight of biomass in L. 153- (intracellular ATP contents).

Comments on the Quality of English Language

Small typos.

Reviewer 2 Report

Comments and Suggestions for Authors

This manuscript describes the production and characterization of a new Bacillus licheniformis strain that was designed to optimize nattokinase production. The amount of the desired nattokinase enzyme produced in batch-fed fermentations by this novel engineered strain was investigated in comparison to another engineered strain, as was the glucose and amino acid consumption of these respective strains, as well as the activity of glycolysis and the TCA cycle and the concentration of ATP in these strains. The optimal glucose feeding and oxygen supply conditions for NK production by the novel engineered strain were also determined here. The manuscript provides adequate justification for the importance of the nattokinase enzyme and for improving its production in liquid fermentation cultures. The methods used in this study were described with sufficient detail to allow this work to be repeated by another laboratory. The findings of this study do advance the understanding of the means of efficiently producing nattokinase in liquid cultures, and would be of interest to the portion of the community studying this enzyme. This reviewer recommends publication of this study once the issues outlined below are addressed.

Minor English style/grammar corrections are needed throughout this manuscript.

Introduction, lines 76-77: It is unclear what is meant by “the most conservative part in the….”. This should be rephrased to clarify its meaning (same wording appears in results section).

Figure 2: As presently formatted, this figure makes it difficult for the reader to make comparisons between the fermentation profiles of the two bacterial strains studied. It would be a more informative figure if the profiles for both strains were shown on the same graph, perhaps with some of the less important parameters omitted. This is also true of figures 6 and 7.

Results, lines 206-208: The statement that glucose and soybean meal consumption was “probably correlated with NK production” requires additional substantiation/explanation. The authors do not explain why the increase in NK production, but loss of PGA production, would result in an increase in glucose consumption, and this is not clear to the reader. The study does show increases in glycolysis and the TCA cycle in the BL11 strain, which could explain the increase in glucose consumption, but again, no explanation is provided as to why the lack of PGA production would upregulate these pathways.

Results, line 211: “increment” should be “increase.”

The authors state that it is unclear whether the measurements of soybean meal concentration in the BL10 fermentation were accurate, but then go on to show this data in Figure 3b. If the authors are unsure of the accuracy of this data, it should not be shown in figure 3.

Comments on the Quality of English Language

Minor English style/grammar corrections are needed throughout this manuscript.
